# The Influence of Selected Environmental Factors on the Number of Somatic Cells in Cistern and Alveolar Milk of Polish Holstein-Friesian Cows

**DOI:** 10.3390/ani14152219

**Published:** 2024-07-31

**Authors:** Piotr Guliński, Michał Kacper Kroszka

**Affiliations:** Faculty of Agricultural Sciences, University in Siedlce, Prusa Street 14, 08-110 Siedlce, Poland

**Keywords:** cows, somatic cells, cisternal and alveolar milk

## Abstract

**Simple Summary:**

The primary aim of the study was to evaluate the variability in the number of somatic cells in cisternal and alveolar milk of Polish Holstein-Friesian cows. The obtained results showed that the SCC and the analyzed characteristics exhibited high variability caused by the influence of various environmental and genetic factors. In reference to cisternal and alveolar milk, inconclusive results were obtained, suggesting that the SCC levels in milk collected from different milking phases were similar.

**Abstract:**

The aim of the study was to evaluate the influence of the milking phase on somatic cell count (SCC) in milk obtained from the cisternal and alveolar parts of udders of selected Polish Holstein-Friesian cows. The study also assessed the impact of other genetic and environmental factors on SCC variability in cisternal and alveolar milk, including: the individual cow, lactation stage, age of cow, production level, milking speed, fat-to-protein ratio, and milking type. The research included 15 cows of Polish Holstein-Friesian breed at different ages, lactation stages, and with varying daily milk yield. A total of 210 milk observations were conducted, including 105 for 1 min milking and 105 for 8 min milking. The results obtained in the study indicated that milk obtained during two different milking phases exhibited similar SCC levels (F for LOGSCC = 0.79). The average actual SCC in milk produced by 15 cows in 105 observations for 1 min milking was 219,000 cells/mL, while for 8 min milking it was 229,000 cells/mL. The results were inconclusive, suggesting that SCC in cisternal and alveolar milk must be influenced by factors other than the milking phase. The analysis of variance conducted for this purpose provided the basis for stating a highly statistically significant effect of the individual cow (F for LOGSCC = 147.9), lactation stage (F for LOGSCC = 54.64), age of cow (F for LOGSCC = 12.39), daily production level (F for LOGSCC = 34.49), milking speed (F for LOGSCC = 17.56), and fat-to-protein ratio (F for LOGSCC = 22.99) on the variability of characteristics defining SCC in milk. In summary, SCC is characterized by high variability, influenced by a range of environmental and genetic factors such as the individual cow, lactation stage, age of cow, milking speed, and dietary fat-to-protein ratio. The influence of milking phase (1 min or 8 min) and milking type (morning or evening) should be considered inconclusive based on the entire population studied. For half of the cows, SCC in cisternal milk was higher than in alveolar milk, while for the other half, the situation was reversed. Further observations are required to confirm the hypothesis regarding the extent to which cows’ immunological response to bacterial infections is concentrated in the cisternal or alveolar part of the udder under national environmental conditions.

## 1. Introduction

Since the development of technology allowing the common assessment of somatic cell count (SCC), information on its level in cow milk has been a fundamental biomarker used to determine cytological quality. Increasing SCC in milk indicates an ongoing inflammatory process within the mammary gland. This situation entails several negative consequences, including reduced milk yield [1,2,3,4,5,6], deterioration of its technological usability [7,8,9], and decreased reproductive efficiency [10,11,12,13,14,15,16]. For these reasons, in dairy plants both domestically and internationally, SCC is considered one of the most important criteria for assessing milk quality in procurement and an essential element in raw milk payment systems.

Information on factors influencing SCC in cow milk over the past 15 years has been the subject of numerous scientific and popular scientific works. Knowledge in this area can be described as extensive, well-documented, and of great applicational and cognitive importance. According to Guliński [17], the use of knowledge regarding SCC in the domestic dairy sector concerns two main entities: dairy plants (determining the quality class and associated milk price) and breeding programs at the stage of assessing the dairy utility of cows. Olechnowicz and Jaśkowski [18] highlighted differences in the upper limit of SCC as a law basis for recognizing milk from healthy udders in various countries worldwide. In the European Union, this limit is 400,000 cells/mL, in Canada 500,000, and in the USA 750,000 cells/mL of milk.

In the specialist literature, the main sources of SCC variability in cow milk, aside from infections, are considered to be: cow age, lactation stage, production season, cattle husbandry technologies, mechanical damage, and indirect causes [9,19]. Referring to environmental factors affecting SCC, Guliński [17] reported that in domestic conditions, lower SCC was found in loose-housing systems with milking parlors and high production levels. Olechnowicz and Jaśkowski [18] identified the main sources of SCC variability in cow milk as: cow age, lactation stage, season, farm size, sampling frequency, metabolic or physiological “stress”, daily milk yield variability, very low or high cow condition score (BCS), genetic differences between udder quarters and cows, and technical error in measurement. In the study by Stocco et al. [20], the greatest sources of SCC variability in milk were also cow age and lactation stage. The authors analyzed 15,936 milk samples from Simmental (9275 samples) and Holstein-Friesian (12,849 samples) cows kept in herds in northern Italy. The analysis of variance results showed that among the various factors analyzed for SCC variability in milk, the highest and statistically significant F-test values were for cow age (F = 583.8**) and lactation stage (F = 648.3**). The significant influence of cow age on SCC in milk was also indicated by Tancin’s research [21], where SCC increased with the number of completed lactations. Thus, the highest SCC was found in milk from cows in their fourth and higher lactations (logarithm of SCC (LOGSCC) = 5.52/mL), while in first lactation cow milk, LOGSCC was 5.22 per mL. LOGSCC in milk from cows in second and third lactations was 5.24 and 5.30, respectively.

Some studies indicate the production season as a factor influencing SCC. In Tancin’s study [21], SCC in milk obtained during the summer was higher compared to milk obtained during the winter. The author reported that in June and July, as many as 18% of analyzed cows had SCC > 1 million per mL. Conversely, Czyżak-Runowska et al. [22] presented a different opinion. Their work showed that bulk milk produced in the autumn–winter period averaged 427,000 somatic cells per mL, while in the spring–summer period, SCC in milk from the same herds was 251,000 cells per mL. Jonas et al. [23] also indicated the negative significant effect of the summer production season, cow age, and lactation stage on SCC in milk. They analyzed 4891 milk samples from Hungarian Holstein cows. The authors found that the highest SCC, expressed in points (LKSP), was found in milk obtained during the summer (5.31), from the oldest cows at five lactations (5.71), and in the last trimester of lactation (5.38). On the contrary, Malinowski [24] suggested that the season of the year did not limit SCC in cow milk. Lambertz et al. [25] highlighted the significant impact of microclimatic conditions in cow barns on SCC in milk. Heat stress caused a decrease in milk yield, percentage fat and protein content, and an increase in SCC. The results of Sadeghi-Sefidmazgi and Rayatdoost-Baghal [26] in the dry climate conditions of Iran indicated that statistically significant factors influencing SCC were: the type of bedding in the stall, milk yield level in lactation, herd size, teat disinfection type, cow maintenance system, and the use of gloves in milking procedures. The highest SCC expressed in points (SCP) was found for cows kept in straw-bedded stalls (SCP = 4.62), producing more than 10,000 kg of milk in lactation (SCP = 3.15), kept in herds < 100 cows (SCP = 3.85), and in herds where gloves were not used in milking procedures (SCP = 3.95).

Barłowska et al. [27] assessed the impact of milking systems on SCC in cow milk. The authors analyzed the importance of four milking systems: manual milking and three types of mechanical milking—bucket, pipeline, and milking parlor—for the cytological quality of bulk milk. The results showed that SCC per mL for the mentioned milking systems was: 108.6, 194.1, 248.5, and 251.9 thousand cells, respectively. In summary, the authors concluded that modern milking systems, such as milking parlors, produce raw material with high microbiological quality but slightly worse cytological quality.

Guliński and Zaremba [28] emphasized the importance of accurately determining SCC in cows evaluated in Poland. These authors compared SCC in cow milk in individual milkings during the month of lactation. The average SCC per mL in milk in the 1st, 2nd, 3rd, and 4th weeks was: 297, 326, 294, and 271 thousand cells, respectively. Analysis of variance for LOGSCC did not show significant differences between different weeks. The low error in assessment confirmed high correlation coefficients between SCC in the 1st, 2nd, 3rd, and 4th lactation weeks, which were: 1, 0.72, 0.80, and 0.74. Based on the obtained results, the authors concluded that over the observed 4-week lactation period, SCC did not change significantly. Consequently, they assessed that the 4-week interval used in official dairy cow utility value assessment accurately reflects actual SCC across four weekly periods. Damm et al. [29] emphasized the need for new methods to diagnose and assess SCC in cow milk. They described the differential somatic cell count (DSCC) as a novel method, accounting for various types of somatic cells. Using flow cytometry, these authors determined the DSCC, the number of polymorphonuclear leukocytes (PMN), and lymphocytes (%) in individual milk samples. The percentage of macrophages was determined by subtracting the percentage of polymorphonuclear leukocytes (PMN), and lymphocytes (%) from 100%. Their results indicated average percentages of lymphocytes, macrophages, and PMN as follows: 5.09%, 33.45%, and 58.68%, respectively. DSCC values ranged widely from 34% to 79% in samples <400,000 cells/mL. Higher DSCC values (53% to 89%) were observed in samples containing >400,000 cells/mL [9]. Halasa et al. [30] demonstrated that increases in SCC in milk correlated significantly with changes in the percentage composition of lymphocytes, macrophages, and PMN. An increase in SCC from <50, 50–100, 100–200, 200–400, to >400 thousand cells/mL of milk corresponded to a decrease in lymphocyte percentage by 26%, 14%, 12%, 11%, and 9%, respectively, a decrease in macrophage percentage by 43%, 35%, 29%, 25%, and 24%, respectively, and an increase in PMN percentage by 31%, 51%, 59%, 64%, and 67%, respectively. Olde Riekerink et al. [31] noted that changes in milk SCC throughout individual milking sessions were mainly attributed to macrophage count.

However, there is a lack of national studies focusing on an exceptionally interesting factor: milk variability in different milking phases. Generally, milk produced between milkings in cow udders is accumulated in the cisternal part (30%) and glandular part (70%) of the udders [32]. Therefore, a decision was made to conduct research aimed primarily at evaluating the influence of milking phase on somatic cell count (SCC) in milk obtained from the cisternal and glandular parts of udders of selected Polish Holstein-Friesian cows. An important additional objective of this study was to analyze the impact of a range of genetic and environmental factors on SCC variability. These factors included: individual cow characteristics, lactation period, cow age, production level, milking rate, fat-to-protein ratio, and milking type.

## 2. Materials and Methods

The study analyzed the impact of selected environmental factors on SCC in milk produced by a selected group of Polish Holstein-Friesian (PHF) cows. This breed belongs to the group of highest-producing cattle breeds and is utilized in 128 countries worldwide [33]. The experiment included a group of 15 cows of varying ages and lactation stages. The study included 3 groups of animals, which, according to official assessments, were characterized by different levels of SCC in milk (>200, 201–400, and >400 thousand/mL). The research was conducted in one of the dairy cattle herds located in Worgule, a part of the administrative structure of Leśna Podlaska commune, Biała Podlaska County, Lublin Voivodeship. The herd consisted of 44 cattle. The animals were kept in a tie-stall system on shallow bedding, and milking was performed using a pipeline milking machine. The feeding system used was partial mixed ration (PMR), consisting of silage, corn silage, beet pulp, and a concentrated feed mixture comprising barley–wheat–rye meal, rapeseed meal, soybean meal, and vitamin premixes. The farm’s area was 25 hectares, with 11 hectares of pasture and the remaining 14 hectares comprising arable land. Four hectares were dedicated to maize cultivation for silage, and grains such as winter barley, winter rye, and spring oats were cultivated on the remaining 10 hectares.

The evaluation of SCC in cow milk was conducted using the DeLaval DCC somatic cell counter. The device operates by counting nuclei of somatic cells stained with a specific propidium iodide reagent. The fluorescence generated was detected using digital imaging, which was then converted using software to count somatic cells. Disposable cassettes were used for the analysis, and SCC was analyzed in less than 1 min. Measurement error of the counter ranged from 7% (for SCC > 1 million cells/mL) to 12% (for SCC < 100,000 cells/mL).

The main objective of the study was to evaluate SCC in milk obtained during different milking phases. For this purpose, milkometers allowing the collection of individual milk samples from any milking phase were used in the milk sampling procedure. Bentley milkometers were employed in the research.

The study was conducted in two stages. In the first stage, conducted between December 2023 and January 2024, the SCC in 1 mL of milk from 15 cows of varying ages and lactation stages was evaluated. Milk samples for evaluation were collected from each cow at two milking phases: 1st and 8th minute during both morning and evening milkings. Milk yield (kg) was also measured at 1st and 8th minute of milking. The duration of each milking session (min.) was recorded. Milk samples from each cow were collected over 7 selected days of the production cycle, ensuring continuity of this period. In total, 210 milk samples were collected (15 × 7 × 2). Milk samples were collected twice a day, at 6 a.m. and 6 p.m.

In the second stage, using breeding documentation and record books, the milk production utility of cows was classified, environmental factors were evaluated, and statistical calculations were performed. The influence of various factors on the variability of SCC in milk was analyzed for individual cows, lactation periods, age groups of animals, designated production groups, cows with different milking rates, animals characterized by a specific fat-to-protein ratio (FPR), and somatic cell count (SCC) in 1 mL of milk. Four lactation periods were distinguished, covering the following days: ≤100, 101–200, 201–300, and >300. Three age groups of cows included animals aged 2 years, 3–4 years, and 5–7 years, respectively. Three daily milk yield levels were considered: <23 kg, 23–30 kg, and >30 kg. Three groups of animals with different milking speed (MS) per minute were identified: ≤1.5 kg, 1.51–2 kg, and >2 kg. Additionally, based on the fat-to-protein ratio (FPR) in milk, the analyzed cow population was classified into three groups: <1.1, 1.1–1.4, and >1.4. Based on the somatic cell count (SCC), animals were grouped into categories: milk containing <200, 200–400, and >400 thousand cells per 1 mL of milk. Since the actual SCC did not meet normal distribution conditions, logarithmic transformation (LOGSCC) was applied in the study. SCC in milk was multiplied by daily milk yield (kg) and transformed into a 10-point logarithmic scale (1–10) (SLSCC). The following modeling approach was used:SLSCC = log10 (daily milk yield × SCC × 1000)(1)

The calculated SLSCC feature allowed for the identification of actual increases or decreases in cell count depending on daily milk yield. For characterizing SCC, the actual SCC was also transformed into a point scale SCS (somatic cell score—SCS). Actual SCC was converted to SCS using the following calculation formula (International Dairy Federation (IDF) [34], Polish Federation of Cattle Breeders and Milk Producers [35]):SCS = log2 (SCC/100,000) + 3(2)

Milk yield in liters after the first and eighth minute of milking was converted to kilograms by multiplying the yield in liters by the specific gravity of milk, 1.028 kg/L.

In statistical calculations, the following linear model was employed:y ijk = μ + Ai + Bj +(AB)ij + eijk(3)
where:

y ijk—somatic cell count;

μ—mean;

Ai—mixed effect of cow (i = 1…15) or lactation period (i = 1…4) or cow age (i = 1, 2, 3) or daily milk yield (i = 1, 2, 3) or SCC range (i = 1, 2, 3) or milking speed (i = 1, 2, 3) or milking type (i = 1, 2) or fat-to-protein ratio (i = 1, 2, 3);

Bj—mixed effect of milking phase (j = 1, 2);

ABij—interaction of milking minute × cow effect or lactation period or cow age or daily milk yield or SCC range or milking rate or milking type or fat-to-protein ratio;

eijk—random error.

Results were subjected to statistical analysis using two-way analysis of variance with the least squares method. Significance of differences between means was assessed using Duncan’s test at *p* ≤ 0.01 level. SAS statistical package procedures FREQ and GLM [36] were used for calculations.

## 3. Results

### 3.1. Population Size and Milk Yield

Table 1 presents data on the distribution of the number of cows used in the experiment. A total of 210 milk observations were conducted, including 105 for the 1 min milking and 105 for the 8 min milking. The study involved 15 cows of varying ages and lactation periods, characterized by different daily milk yields. The highest percentage was attributed to cows in the first trimester of lactation (38.1%), aged 3–4 years (53.3%), and with a daily milk yield exceeding 30 kg (42.9%). Within the other designated factors such as SCC class contribution in milk, milk let-down speed, and FPR, the highest proportion of milk samples analyzed concerned animals with the lowest SCC (<200 thousand/mL), milk let-down speeds between 1.51 to 2 kg/min, and optimal FPR size in milk of 1.1 to 1.4. The percentage distribution of cows in these subcategories of factors was respectively: 73.3, 42.9, and 53.3%.

The average milk yield per single milking for cows at 1 and 8 min of milking was 3 kg and 14.2 kg respectively (Table 1). Individual cows showed significant variation in this characteristic. Cow number 1 exhibited the highest yield after 1 min of milking (4.9 kg), while cow number 7 had the lowest (1.6 kg). Similar high variation was observed in milk yield after 8 min of milking. The highest milk yield was found in cow number 15 (17.9 kg) and the lowest in cow number 6 (10.3 kg). Milk yields for the remaining cows ranged between these extreme values.

Evaluation of milk yield within the remaining factors indicated that animals in the first trimester of lactation, aged 5–7 years, with the lowest SCC in milk (<200 thousand/mL), the highest milk let-down speed (>2 kg/min), and the highest FPR size in milk (>1.4) demonstrated the highest values of this indicator. Specifically, for 1 min of milking: 3.8, 3.4, 3.6, 3.2, 3.9, and 3.9 kg; for 8 min: 17.2, 16.6, 17.2, 15.0, 18.2, and 15.9 kg. The study showed that milk yields after 1 and 8 min in morning and evening milkings exhibited similar and statistically nonsignificant values. Morning milk yields after 1 and 8 min differed compared to evening milk yields (3.1 vs. 2.9 and 14.7 vs. 13.8), but statistical analysis did not confirm the significance of these differences at *p* ≤ 0.05.

### 3.2. Analysis of Variance Results for Somatic Cell Counts

The primary goal of the study was to assess the impact of various environmental and genetic factors on the variability of actual SCC, LOGSCC, SLSCC, and SCP. The results presented in Table 2 provided the basis for confirming the hypothesis of a highly statistically significant effect of the cow (F for LOGSCC = 147.9), lactation period (F for LOGSCC = 54.64), cow age (F for LOGSCC = 12.39), daily yield level (F for LOGSCC = 34.49), milk let-down speed (F for LOGSCC = 17.56), and fat-to-protein ratio (F for LOGSCC = 22.99) on the characteristics of somatic cell counts in milk in the population of cows studied. These characteristics obtained high F-test values, indicating a strong influence of individual factors and their subclasses on the somatic cell counts and three additional characteristics assessed in the study.

The F-test results indicated that the highest variability in somatic cell counts (F for LOGSCC = 271.67) was observed in milk from cows belonging to three classes of differentiated SCC per 1 mL. This result should be considered obvious and derived from the nature of SCC variation in milk depending on the intensity of udder inflammatory states in cows.

One of the primary goals of the study and the experiment conducted was to investigate whether milk obtained from different phases of milking, i.e., alveolar milk and cisternal milk, exhibited similar SCC levels. As indicated in the methodology, milk samples from each cow were collected twice during each milking, i.e., after 1 and after 8 min of milking. Based on the data in Table 2, it should be noted that in the studied population of cows, milk obtained in two different milking phases exhibited similar SCC levels (F for LOGSCC = 0.79). Similarly, the F-test values for the interaction of milking phase with all analyzed factors did not prove statistically significant sources of variability.

### 3.3. Evaluation of the Influence of Other Analyzed Factors on SCC

The main objective of the study was to verify the null hypothesis assuming differences in SCC in milk obtained in the initial and final phases of milking, i.e., from the cisternal and alveolar parts of cows’ udders. Therefore, detailed data on the variability of SCC and the influence of the factors specified in the methodology on SCC and its other characteristics are presented in Table 3 and Table 4. The results obtained showed that the actual SCC in milk produced by 15 cows in 105 observations after 1 min of milking was 219 thousand/mL and after 8 min it was 229 thousand/mL. Both cisternal and alveolar milk from individual cows were characterized by high variability in SCC per 1 mL (Table 3). For cisternal milk, the coefficient of variation for SCC ranged from 16 to 67%, and for alveolar milk from 17 to 67%. The analysis of the actual variance of SCC after logarithmic transformation (LOGSCC) showed that in the entire population, the difference in SCC in milk obtained after 1 (LOGSCC = 4.3) and 8 min (LOGSCC = 4.3) of milking was not statistically significant at *p* ≤ 0.05. Likewise, the difference in average values of SCP (2.6 vs. 2.5) in cisternal and alveolar milk was not statistically significant at *p* ≤ 0.05. Data analysis in Table 4 showed that the actual SCC in milk after 1 min compared to 8 min of milking was higher in seven cows (cows 1, 3, 4, 5, 9, 10, 11, and 12), lower in seven cows (cows 2, 6, 7, 8, 12, 13, 14, and 15), and the same in one cow (cow 4). The analysis of variance confirmed the significance of differences in SCC after logarithmic transformation (LOGSCC) between cisternal and alveolar milk in the case of the first group of cows (those with higher SCC in cisternal milk): cows 4, 5, 8, 11, and in the case of the second group of cows (those with lower SCC in cisternal milk): cows 2, 6, 8, 15.

The results presented in Table 2 must be considered ambiguous and allow for the conclusion that SCC in cisternal and alveolar milk must be determined via factors other than milking phase. Therefore, data on the influence of lactation period, cow age, milk yield, udder health status (SCC classes), milk let-down speed, fat-to-protein ratio, and milking type (morning or evening) on somatic cell counts in cisternal and alveolar milk of the studied cow population are presented in Table 4. These data indicate that factors such as lactation period, cow age, milk yield, udder health status, milk let-down speed, and the level of energy-protein balance in feed doses (FPR) significantly and statistically differentiated SCC in cisternal and alveolar milk obtained from the studied cow population. The highest SCC per 1 mL was found in milk from cows in the last period of lactation (SCC = 549 thousand), aged 3–4 years (SCC = 279 thousand), producing 23.1–30 kg of milk per day (401 thousand), with milking speeds between 1.51–2 kg (SCC = 285 thousand), FPR ≤ 1.1 (SCC = 454 thousand). The lowest SCC was noted in milk from cows in the third trimester of lactation (SCC = 30 thousand), aged 2 years (SCC = 72 thousand), producing over 30 kg of milk per day (44 thousand), with milking speeds exceeding 2 kg (SCC = 43 thousand), and FPR > 1.4 (SCC = 46). SCC values for the remaining subclasses of individual factors were intermediate relative to the above boundary values. As already noted in Section 3.2, the analysis of variance confirmed the significance of differences in SCC for the distinguished subclasses of main factors, such as lactation period, cow age, production level, SCC, and FPR at *p* ≤ 0.05. For the results of the study, particular importance was attributed to the assessment of the impact of these factors on SCC in cisternal milk (obtained after 1 min of milking) and in alveolar milk (obtained after 8 min of milking). The data in Table 4 showed that the nature of interaction of the main factors determined in the work methodology on SCC variability at 1 and 8 min of milking was identical to that described above for full milking milk. Separate commentary is required for the data concerning changes in SCC depending on the health status of cow udders, determined by SCC class. The demonstrated increase in SCC in milk at 1 and 8 min of milking with deteriorating udder health status (increase in SCC class) should be considered obvious and resulting from the adopted principles of assessing the interdependence between SCC and udder health status in cows. Of greater interest for the study results is to evaluate the observed lack of statistically significant differences in SCC in milk at 1 and 8 min of milking in cows with varied udder health status. The conducted research showed that SCC in foremilk and hindmilk within individual classes of its quality formed at a similar level. This result indicates that in the studied population of cows, the health status of cows differentially affected the animals’ immune response in a statistically significant manner, but to the same extent in both cisternal and alveolar milk. The results regarding the effect of milking speed (MS) on SCC in the evaluated population of cows should also be noted. The results presented in Table 4 showed that in the authors’ own research, with an increase in MS, SCC decreased, both in cisternal and alveolar milk. The lowest SCC characterized the milk of cows with MS > 2 kg (SCC = 43 thousand), while in the milk of peers with the lowest MS < 1.5 kg, the average SCC was 275 thousand.

## 4. Discussion 

Discussion of the obtained results is complicated due to the small number of scientific papers directly related to the issues discussed in the work Results similar to those obtained in autors own work were described by Sølverød et al. [37]. These authors studied SCC in cisternal and alveolar milk in seven cows of the Norwegian Red breed.The results obtained showed that most cows had higher SCC in alveolar milk (four pieces). The aim of the research conducted by Sarikaya et al. [38] was to assess the level of immune response in different quarters of dairy cows. For this purpose, the authors analyzed the composition and structure of the leukocyte population in milk obtained at different milking phases. The assessment covered nine dairy cows from which milk was collected separately from each quarter, assuming that SCC could not exceed 125 thousand/1 mL. The milk collected, depending on the milking phase, was divided into: cisternal milk (C), alveolar milk 0–25%, 25–50%, 50–75%, 75–100% (respectively A25; A50; A75; A100) and residual milk (R). Each batch of milk was analyzed for major chemical components, SCC, and the proportion and structure of leukocyte populations. The results of the study showed that fat content steadily increased during milking and reached its highest values in R milk. Protein and lactose increased from C to A25, decreased from A25 to A100, and reached a minimum in R. Levels of Na and Cl ions decreased from C to A25, then increased from A50 to R. SCC in cisternal milk reached a high level, then in alveolar milk its level decreased, reaching a minimum in A25 alveolar milk. The highest SCC was found in residual milk (R). The structure of the somatic cell population in different milk batches was characterized by an inverse trend in the case of macrophages (M) and polymorphonuclear neutrophils (PMN). The proportion of M was highest in cisternal milk (C), while the proportion of PMN increased regularly in individual milk batches, reaching its maximum in residual milk (R). The proportion of lymphocytes (L) changed similarly to PMN in milk C, A25, A50, A75, and R. However, the highest proportion of L was found in alveolar milk from the last phase A100. The proportions of L, PMN, and M were 9.3%, 38.2%, and 52.3% in cisternal (C), 10.9%, 64%, and 25.1% in A25-A100 in alveolar milk, and 10.2%, 64.9%, and 24.8% in residual milk (R). The study found that in healthy quarters, M were the dominant type of somatic cells in foremilk (C). They are located near the streak canal, the main entry point for pathogens. They constitute the first immunological barrier to attacking pathogens. In contrast, in alveolar, the dominant fraction of somatic cells was PMN. Since each fraction of somatic cells had a higher percentage share in cisternal than in early phases of alveolar milk, the work summary concluded that this fact indicates the key role of immunological defense in the milk sinuses of cow udders.

Sarikaya et al. [39] in their study investigated the influence of SCC and milk obtained at different phases of milking on its composition, somatic cell population distribution, and mRNA expression related to varying levels of mammary gland inflammation in cows. Three types of milk samples were distinguished as follows: cisternal milk (C), first 400 g of alveolar milk (A1), and remaining alveolar milk (A2). Based on SCC, milk samples were categorized into four groups: <12 × 10^3^/mL, 12 to 100 × 10^3^/mL, 100 to 350 × 10^3^/mL, and >350 × 10^3^/mL. The following milk characteristics were analyzed: SCC, fat, protein, lactose, sodium, potassium, and electrical conductivity. Somatic cell populations were classified as lymphocytes (L), macrophages (M), and neutrophils (PMN). The SCC decreased from the highest level in C to the lowest in A1, then increased in A2 across all groups. Lactose decreased with increasing SCC but remained unchanged during milking. Sodium and chloride concentrations, as well as electrical conductivity, increased with SCC levels, being higher in C compared to A1 and A2. Milk fraction and SCC had no impact on protein content. Milk fraction and SCC significantly influenced leukocyte distribution; lymphocytes predominated in Group 1, with lower percentages in Groups 2, 3, and 4. Macrophages were most prevalent in Group 2, decreasing in Groups 3 and 4, while PMN increased from Group 2 to 4. During milking, macrophage proportions decreased across all groups, whereas PMN proportions increased. The authors found that milk batch did not affect lymphocyte proportions. mRNA expression in all inflammatory states increased with SCC levels. In summary, the authors concluded that milking phase, associated milk batch, and SCC class significantly influenced the structure and proportions of leukocyte types. They noted the surprising high lymphocyte content and low inflammatory mRNA expression in milk with SCC < 12 × 10^3^/mL, suggesting a potentially altered and decreased immune response readiness to pathogen attack.

Tančin et al. [40] evaluated the relationship between milking parameters and SCC in milk from 22 Holstein cows. SCC was assessed in morning milkings. Parameters related to milking speed (milking onset delay, duration of decline phase, and duration of extension phase) for individual quarters and the entire udder were evaluated daily. Results indicated that both quarter and total milk yield decline was associated with increased milk SCC. Milking characteristics such as lower peak yield and longer milking extension phase were observed in quarters with high SCC (>500,000 cells/mL) compared to those with low SCC (<200,000 cells/mL). The duration of the decline phase for each quarter affected all measured parameters except the duration of the growth phase. Quarters with longer decline phases (> or =80 s) showed higher SCC and peak milk yield during milking. These findings are consistent with the authors’ own results, where an increase in SCC was accompanied by a decline in milking speed, indicating an improvement in udder health. Similar results were obtained by Mijić et al. [41]. Their study aimed to examine the relationship between milking speed and SCC in milk from healthy Holstein cows. Research on 220 Holstein-Friesian cows showed that a minimal SCC level expressed in points (MS = 2.78 to 3.81) occurred in cows producing maximum milk release speed, defined in cited studies as 2.7 to 4.5 kg/min. The authors attributed this correlation to the width of the teat canal and the strength of the teat sphincter, which prevent invasion by pathogenic microorganisms into the udder. The correlation coefficient between MS and maximum milk flow in the first and second lactations was positive (r = 0.06 and 0.14), while in the third lactation, it was statistically significant but negative (r = −0.07*). The highest positive correlation, statistically significant, was found between the average milking speed and MS in the first lactation group (r = 0.40**). In another study by Mijić et al. [42], it was found that SCC in Holstein cow milk (MS = 3.03) was lowest when the length of the milk influence phase increased from 0.3 to 0.6 min, whereas in Simmental cows (MS = 2.96), it was from 0.6 to 0.9 min. Prolongation of the plateau phase (dPP) had a beneficial effect on udder health in both breeds (MS = 2.46 and 2.50).

## 5. Conclusions

The obtained results allowed to conclude that generally, within the studied population of cows, milk obtained in two different milking phases showed similar levels of SCC (F-test for LOGSCC = 0.79). The average actual SCC in milk produced by 15 cows in 105 observations at 1 min of milking was 219 thousand/mL, and at 8 min it was 229 thousand/mL. Detailed evaluation of data regarding SCC in cisternal milk and alveolar milk of individual cows revealed that the actual SCC at 1 min compared to 8 min of milking was higher in seven cows (cows no. 1, 3, 4, 5, 9, 10, 11, and 12), lower in seven cows (cows no. 2, 6, 7, 8, 12, 13, 14, and 15), and the same in one cow (cow no. 4). Analysis of variance confirmed the significance of differences in SCC after logarithmic transformation (LOGSCC) between cisternal and alveolar milk in Group I cows (having higher SCC in cisternal milk) for cows no. 4, 5, 8, and 11, and in Group II cows (having lower SCC in alveolar milk) for cows no. 2, 6, 8, and 15. The results obtained must be considered ambiguous and suggest that SCC in cisternal and alveolar milk must be influenced by factors other than milking phase. Therefore, the study evaluated the impact of various genetic and environmental factors on SCC and other characteristics in milk. The results of the analysis of variance provided evidence of highly statistically significant effects of individual cow, lactation period, age of animal, daily milk yield, milking rate, and fat-to-protein ratio on the variability of characteristics defining somatic cell count in the studied population of cows. Based on the obtained results, the following conclusions were formulated: SCC is characterized by high variability. In the authors’ own research, it was influenced by a number of environmental and genetic factors including: individual cow, lactation period, age of cow, milking rate, and fat-to-protein ratio. The impact of factors such as milking phase (1 min or 8 min) and milking type (morning or evening) should be considered as inconclusive based on the entire population studied. For half of the cows, SCC in foremilk was higher than in hindmilk, while for the other half the situation was reversed. Confirming the hypothesis of how cows’ immune response to bacterial infections is concentrated in the cisternal and alveolar milk, under national environmental conditions, requires further observation.

## Figures and Tables

**Table 1 animals-14-02219-t001:** The influence of the analyzed factors on the actual milk yield in a single milking (kg).

Factor	Milking Phase			
	First Minute		Eighth Minute	
	Observation Number N (%)	Milk Yield in Single Milking (kg) LSM (±SD)	Observation Number N (%)	Milk Yield in Single Milking (kg) LSM (±SD)
Cow (nr)				
1–15	15 × 7 (6.7)	4.9 ^A^ (±0.2)	15 × 7 (6.7)	16.7 ^B^ (±1.1)
Lactation period/days/				
≤100	40 (38.1)	3.8 ^A^ (±0.8)	40 (38.1)	17.2 ^B^ (±1.6)
101–200	23 (21.9)	2.5 ^A^ (±0.5)	23 (21.9)	14.2 ^B^ (±1.4)
201–300	14 (13.3)	2.7 ^A^ (±0.2)	14 (13.3)	11.9 ^B^ (±0.8)
>300	28 (26.7)	2.5 ^A^ (±0.5)	28 (26.7)	11.1 ^B^ (±1.2)
Age of cows/years/				
≤2	14 (13.3)	3.4 ^A^ (±1.6)	14 (13.3)	14.7 ^B^ (±2.2)
3–4	56 (53.3)	2.7 ^A^ (±0.8)	56 (53.3)	12.6 ^B^ (±2.3)
5–7	35 (33.3)	3.4 ^A^ (±0.5)	35 (33.3)	16.6 ^B^ (±2.4)
Daily milk yield/kg/				
≤23	30 (28.6)	2.5 ^A^ (±0.5)	30 (28.6)	10.8 ^B^ (±0.7)
24–30	30 (28.6)	2.6 ^A^ (±0.6)	30 (28.6)	13.2 ^B^ (±0.8)
>30	45 (42.9)	3.6 ^A^ (±0.9)	45 (42.9)	17.2 ^B^ (±1.4)
Somatic cell count/thous/1 mL/				
≤200	77 (73.3)	3.2 ^A^ (±0.9)	77 (73.3)	15.0 ^B^ (±2.8)
201–400	14 (13.3)	2.9 ^A^ (±0.2)	14 (13.3)	11.6 ^B^ (±1.4)
>400	14 (13.3)	2.1 ^A^ (±0.4)	14 (13.3)	12.2 ^B^ (±1.8)
Milking speed (kg/min.)				
≤1.5	35 (33.3)	2.5 ^A^ (±0.5)	35 (33.3)	10.9 ^B^ (±0.9)
1.51–2	45 (42.9)	2.9 ^A^ (±0.9)	45 (42.9)	14.5 ^B^ (±1.3)
>2	25 (23.8)	3.9 ^A^ (±0.7)	25 (23.8)	18.2 ^B^ (±1.3)
Fat/protein ratio in milk				
≤1.1	14 (13.3)	2.7 ^A^ (±0.1)	14 (13.3)	12.6 ^B^ (±0.6)
1.11–1.4	56 (53.3)	2.5 ^A^ (±0.6)	56 (53.3)	13.5 ^B^ (±3.1)
>1.4	35 (33.3)	3.9 ^A^ (±0.9)	35 (33.3)	15.9 ^B^ (±2.5)
Milking type				
morning	45 (42.9)	3.1 ^A^ (±0.9)	45 (42.9)	14.7 ^B^ (±3.1)
evening	60 (57.1)	2.9 ^A^ (±0.9)	60 (57.1)	13.8 ^B^ (±2.8)
Total	105 (100)	3.0 ^A^ (±0.9)	105 (100)	14.2 ^B^ (±2.9)

Means in rows marked with different letters differ significantly at *p* ≤ 0.05.

**Table 2 animals-14-02219-t002:** Results of variance analysis for SCC, SLSCC, and SCP (F-test values) for the factors of variability determined in the methodology of the work.

Effect	Characteristics for the Number of Somatic Cells		
	LOGSCC	SLSCC	SCP
	F-Test Value		
Cow (nr)	147.9 **	128.42 **	147.9 **
Lactation period (days)	54.64 **	43.21 **	54.64 **
Age of cows (years)	12.39 **	8.06 **	12.39 **
Daily milk yield (kg)	34.49 **	21.21 **	34.49 **
Somatic cell counts class	271.67 **	232.67 **	271.67 **
Milking speed	17.56 **	9.98 **	17.56 **
Fat/protein ratio	22.99 **	15.40 **	22.99 *
Milking type	0.21	0.07	0.21
Milking minute	0.79	564.85 **	0.79
Milking minute × cow	1.51	1.60	1.51
Milking minute × lactation period	0.56	0.76	0.56
Milking minute × age of cow	0.07	0.15	0.07
Milking minute × daily milk yield	0.08	0.08	0.08
Milking minute × SCC class	0.38	0.91	0.38
Milking minute × milking speed	0.14	0.03	0.14
Milking minute × fat/protein ratio	0.14	0.37	0.14
Milking minute × milking type	0.04	0.02	0.04

*—the effect of the factor is statistically significant at *p* ≤ 0.05. **—the effect of the factor is highly statistically significant at *p* ≤ 0.01.

**Table 3 animals-14-02219-t003:** Variability of the actual number of somatic cells in cisternal and alveolar milk of individual cows included in the experiment.

Cow nr	Observation Number n	Actual Somatic Cell Counts			Observation Number	Actual Somatic Cell Counts		
		Average	Min/Max	V (%)		Average	Min/Max	V (%)
	Cisternal Milk				Alveolar Milk			
1	7	85	25/163	67	7	76	32/160	67
2	7	69	39/88	25	7	90	53/112	28
3	7	38	14/65	52	7	20	9/28	39
4	7	13	7/19	31	7	13	4/16	36
5	7	76	13/120	55	7	49	27/73	45
6	7	77	48/97	23	7	82	58/97	17
7	7	848	590/1016	21	7	869	525/1028	25
8	7	333	220/539	34	7	431	301/609	28
9	7	39	28/61	32	7	21	17/29	21
10	7	37	13/71	57	7	28	17/43	36
11	7	53	37/63	16	7	33	22/39	19
12	7	891	377/1443	43	7	864	334/1401	48
13	7	53	34/64	23	7	57	44/65	16
14	7	666	225/824	33	7	792	362/1152	32
15	7	7	1/10	54	7	9	6/18	51

**Table 4 animals-14-02219-t004:** The impact of the milking phase and other factors on the somatic cell count (SCC), the decimal logarithm of their actual number (LOGSCC), the somatic cell count expressed on a logarithmic scale (SLSCC), and the somatic cell count expressed on a score scale (SCP) per 1 mL of milk.

Cow /nr/	Milking Phase								Complete Milking			
	First Minute				Eighth Minute							
	SCC /thous./	LOGSCC	SLSCC	SCP/pts./	SCC /thous./	LOGSCC	SLSCC	SCP /pts./	SCC /thous./	LOGSCC	SLSCC	SCP /pts./
	LSM (±SD)				LSM (±SD)				LSM (±SD)			
Cow/nr/												
1	85 ^A^ (±57)	4.2 ^A^ (±0.8)	5.5 ^A^ (±0.3)	2.4 ^A^ (±1.2)	76 ^A^ (±51)	4.2 ^A^ (±0.6)	6.0 ^A^ (±0.3)	2.4 ^A^ (±0.9)	81 ^4^ (±52)	4.2 ^3, 4^ (±0.7)	5.8 ^3, 4^ (±0.4)	2.4 ^3, 4^ (±1.0)
2	69 ^A^ (±17)	4.2 ^A^ (±0.3)	5.5 ^A^ (±0.1)	2.4 ^A^ (±0.4)	91 ^A^ (±26)	4.5 ^B^ (±0.3)	6.1 ^B^ (±0.1)	2.8 ^B^ (±0.5)	80 ^4^ (±24)	4.3 ^3^ (±0.3)	5.8 ^3^ (±0.4)	2.6 ^3^ (±0.5)
3	39 ^A^ (±8)	3.5 ^A^ (±0.6)	4.9 ^A^ (±0.2)	1.4 ^A^ (±0.9)	21 ^B^± (4)	2.9 ^B^ (±0.5)	5.3 ^A^ (±0.2)	0.6 ^B^ (±0.7)	30 ^4^ (±17)	3.2 ^7^ (±0.6)	5.1 ^8^ (±0.3)	1.0 ^7^ (±0.9)
4	13 ^A^ (±4)	2.5 ^A^ (±0.3)	4.6 ^A^ (±0.2)	0.1 ^A^ (±0.5)	13 ^A^ (±5)	2.5 ^A^ (±0.5)	5.3 ^B^ (±0.2)	−0.1 ^A^ (±0.7)	13 ^4^ (±4)	2.5 ^8^ (±0.4)	4.9 ^9^ (±0.4)	−0.1 ^8^ (±0.6)
5	76 ^A^ (±42)	4.1 ^A^ (±0.9)	5.0 ^A^ (±0.4)	2.2 ^A^ (±1.3)	49 ^B^ (±22)	3.8 ^B^ (±0.5)	5.8 ^B^ (±0.2)	1.9 ^B^ (±0.7)	62 ^4^ (±35)	3.9 ^4, 5^ (±0.7)	5.4 ^6^ (±0.5)	2.0 ^4, 5^ (±1.0)
6	77 ^A^ (±18)	4.3 ^A^ (±0.3)	5.3 ^A^ (±0.1)	2.6 ^A^ (±0.4)	82 ^A^ (±14)	4.4 ^A^ (±0.2)	5.9 ^B^ (±0.1)	2.7 ^A^ (±0.3)	80 ^4^ (±15)	4.4 ^3^ (±0.2)	5.6 ^5^ (±0.3)	2.6 ^3^ (±0.3)
7	849 ^A^ (±175)	6.7 ^A^ (±0.2)	6.1 ^A^ (±0.1)	6.1 ^A^ (±0.3)	870 ^A^ (±224)	6.7 ^A^ (±0.3)	6.9 ^B^ (±0.1)	6.1 ^A^ (±0.4)	859 ^1^ (±194)	6.7 ^1^ (±0.3)	6.5 ^1^ (±0.4)	6.1 ^1^ (±0.4)
8	334 ^A^ (±112)	5.8 ^A^ (±0.3)	5.9 ^A^ (±0.1)	4.7 ^A^± (0.4)	431 ^A^ (±123)	6.1 ^B^ (±0.3)	6.6 ^B^ (±0.1)	5.1 ^B^ (±0.4)	382 ^3^ (±124)	5.9 ^2^ (±0.3)	6.3 ^2^ (±0.4)	4.9 ^2^ (±0.5)
9	39 ^A^ (±12)	3.6 ^A^ (±0.3)	5.0 ^A^ (±0.1)	1.6 ^A^ (±0.4)	22 ^B^ (±4)	3.1 ^B^ (±0.2)	5.4 ^B^ (±0.1)	0.8 ^B^ (±0.3)	30 ^4^ (±12)	3.3 ^7^ (±0.4)	5.2 ^7, 8^ (±0.2)	1.2 ^7^ (±0.6)
10	37 ^A^ (±21)	3.5 ^A^ (±0.6)	5.0 ^A^ (±0.3)	1.4 ^A^ (±0.8)	29 ^A^ (±10)	3.3 ^A^ (±0.4)	5.7 ^B^ (±0.1)	1.1 ^A^ (±0.5)	33 ^4^ (±16)	3.4 ^7^ (±0.5)	5.3 ^7^ (±0.4)	1.2 ^7^ (±0.7)
11	53 ^A^ (±8)	3.9 ^A^ (±0.2)	5.1 ^A^ (±0.1)	2.1 ^A^ (±0.5)	33 ^B^ (±6)	3.5 ^B^ (±0.2)	5.7 ^B^ (±0.1)	1.4 ^B^ (±0.2)	43 ^4^ (±12)	3.7 ^5, 6^ (±0.3)	5.4 ^6^ (±0.3)	1.7 ^5, 6^ (±0.4)
12	891 ^A^ (±383)	6.7 ^A^ (±0.5)	6.3 ^A^ (±0.2)	6.0 ^A^ (±0.7)	865 ^A^ (±419)	6.6 ^A^ (±0.6)	6.9 ^A^ (±0.2)	5.9 ^A^ (±0.9)	878 ^1^ (±385)	6.7 ^1^ (±0.5)	6.7 ^1^ (±0.4)	5.9 ^1^ (±0.8)
13	53 ^A^ (±11)	3.9 ^A^ (±0.2)	5.2 ^A^ (±0.1)	2.0 ^A^ (±0.3)	57 ^A^ (±9)	4.0 ^A^ (±0.2)	6.0 ^B^ (±0.1)	2.2 ^A^ (±0.2)	55 ^4^ (±10)	3.9 ^3, 4, 5^ (±0.2)	5.6 ^4, 5^ (±0.4)	2.1 ^3, 4, 5^ (±0.3)
14	666 ^A^ (±218)	6.4 ^A^ (±0.5)	6.2 ^A^ (±0.2)	5.6 ^A^ (±0.7)	793 ^A^ (±254)	6.6 ^A^ (±0.4)	7.0 ^B^ (±0.1)	5.9 ^A^ (±0.5)	729 ^2^ (±237)	6.5 ^1^ (±0.4)	6.6 ^1^ (±0.5)	5.8 ^1^ (±0.6)
15	7 ^A^ (± 3)	1.7 ^A^ (±0.9)	4.3 ^A^ (±0.4)	−1.3 ^A^ (±1.2)	9 ^A^ (±5)	2.1 ^B^ (±0.4)	5.2 ^B^ (±0.2)	−0.6 ^B^ (±0.6)	8 ^4^ (±4)	1.9 ^9^ (±0.7)	4.8 ^9^ (±0.5)	−0.9 ^9^ (±0.9)
Lactation period/days/												
≤100	51 ^3^ (±36)	3.6 ^3^ (±1.1)	5.1 ^2^ (±0.5)	1.5 ^3^ (±1.5)	49 ^3^ (±37)	3.6 ^2^ (±0.9)	5.8 ^2^ (±0.4)	1.5 ^3^ (±1.3)	50 ^3^ (±36)	3.6 ^3^ (±3.6)	5.5 ^2^ (±0.5)	1.5 ^3^ (±1.4)
101–200	234 ^2^ (±315)	4.3 ^2^ (±1.7)	5.3 ^2^ (±0.7)	2.6 ^2^ (±2.4)	263 ^2^ (±382)	4.2 ^2^ (±1.8)	5.9 ^2^ (±0.7)	2.5 ^2^ (±2.5)	248 ^2^ (±346)	4.3 ^2^ (±1.7)	5.6 ^2^ (±0.8)	2.5 ^2^ (±2.4)
201–300	39 ^3^ (1±6)	3.6 ^3^ (±0.5)	4.9 ^3^ (±0.2)	1.5 ^3^ (±0.7)	21 ^3^ (±6.3)	3.0 ^3^ (±0.4)	5.4 ^3^ (±0.2)	0.7 ^3^ (±0.5)	30 ^3^ (±15)	3.3 ^3^ (±0.5)	5.2 ^3^ (±0.3)	1.1 ^3^ (±0.7)
>300	537 ^1^ (±406)	5.9 ^1^ (±1.1)	5.9 ^1^ (±0.4)	4.8 ^1^ (±1.5)	562 ^1^ (±407)	5.9 ^1^ (±1.0)	6.6 ^1^ (±0.5)	4.9 ^A^ (±1.5)	549 ^1^ (±403)	5.9 ^1^ (±1.1)	6.3 ^1^ (±0.6)	4.9 ^1^ (±1.5)
Age of cow/years/												
2	81 ^2^ (±48)	4.1 ^2^ (±0.8)	5.3 ^1, 2^ (0±.4)	2.3 ^2^ (±1.2)	63 ^2^ (±40)	3.9 ^2^ (±0.6)	5.9 ^1^ (±0.3)	2.1 ^2^ (±0.8)	72 ^3^ (±44)	4.1 ^2^ (±0.7)	5.6 ^2^ (±0.5)	2.2 ^2^ (±1.0)
3–4	266 ^1^ (±320)	4.8 ^1^ (±1.3)	5.5 ^1^ (±0.5)	3.3 ^1^ (±1.8)	293 ^1^ (±359)	4.7 ^1^ (±1.5)	6.1 ^1^ (±0.6)	3.2 ^1^ (±2.2)	279 ^1^ (±339)	4.8 ^1^ (±1.4)	5.8 ^1^ (±0.7)	3.2 ^1^ (±2.0)
5–7	200 ^1, 2^ (±386)	3.7 ^2^ (±1.8)	5.1 ^2^ (±0.7)	1.6 ^2^ (±2.6)	195 ^1, 2^ (±383)	3.7 ^2^ (±1.7)	5.8 ^1^ (±0.7)	1.7 ^2^ (±2.4)	197 ^1, 2^ (±382)	3.7 ^2^ (±1.7)	5.5 ^2^ (±0.8)	1.7 ^2^ (±2.5)
Daily milk yield/kg/												
≤23	305 ^1^ (±340)	4.9 ^1^ (±1.4)	5.5 ^1^ (±0.5)	3.5 ^1^ (±1.9)	329 ^1^ (±361)	4.9 ^1^ (±1.5)	6.2 ^1^ (±0.7)	3.4 ^1^ (±2.2)	317 ^1^ (±348)	4.9 ^1^ (±1.4)	5.8 ^1^ (±0.6)	3.5 ^1^ (±2.1)
23.1–30	393 ^1^ (±425)	5.1 ^1^ (±1.5)	5.6 ^1^ (±0.6)	3.7 ^1^ (±2.2)	409 ^1^ (±458)	4.9 ^1^ (±1.8)	6.3 ^1^ (±0.8)	3.5 ^1^ (±2.5)	401 ^1^ (±438)	5.0 ^1^ (±1.6)	5.9 ^1^ (±0.8)	3.6 ^1^ (±2.4)
>30	46 ^2^ (±36)	3.4 ^2^ (±1.1)	5.0 ^2^ (±0.5)	1.3 ^2^ (±1.5)	43 ^2^ (±34)	3.4 ^2^ (±0.9)	5.7 ^1^ (±0.4)	1.3 ^2^ (±1.2)	44 ^2^ (±35)	3.4 ^2^ (±0.9)	5.4 ^2^ (±0.5)	1.3 ^2^ (±1.4)
Somatic cell count class/thous./1 mL/												
≤200	49 ^3^ (±34)	3.6 ^2^ (±0.9)	5.1 ^2^ (±0.4)	1.5 ^1^ (±1.3)	44 ^3^ (±33)	3.5 ^2^ (±0.8)	5.7 ^1^ (±0.4)	1.4 ^2^ (±1.2)	46 ^3^ (±33)	3.5 ^2^ (±0.9)	5.4 ^2^ (±0.5)	1.5 ^2^ (±1.3)
201–400	612 ^2^ (±395)	6.2 ^1^ (±0.6)	6.2 ^1^ (±0.2)	5.3 ^1^ (±0.9)	647 ^2^ (±372)	6.3 ^1^ (±0.5)	6.8 ^1^ (±0.3)	5.5 ^1^ (±0.8)	630 ^2^ (±377)	6.3 ^1^ (±0.6)	6.5 ^1^ (±0.4)	5.4 ^1^ (±0.8)
>400	757 ^1^ (±212)	6.6 ^1^ (±0.4)	6.2 ^1^ (±0.2)	5.8 ^1^ (±0.6)	831 ^1^ (±233)	6.7 ^1^ (±0.3)	6.9 ^1^ (±0.1)	6.0 ^1^ (±0.5)	794 ^1^ (±222)	6.6 ^1^ (±0.4)	6.6 ^1^ (±0.4)	5.9 ^1^ (±0.5)
Milking speed/kg/min./												
≤1.5	265 ^1^ (±329)	4.7 ^1^ (±1.4)	5.4 ^1^ (±0.6)	3.2 ^1^ (±2.0)	286 ^1^ (±350)	4.7 ^1^ (±1.6)	6.1 ^1^ (±0.7)	3.1 ^1^ (±2.2)	275 ^1^ (±337)	4.7 ^1^ (±1.5)	5.8 ^1^ (±0.7)	3.1 ^1^ (±2.1)
1.51–2	281 ^1^ (±380)	4.7 ^1^ (±1.5)	5.5 ^1^ (±0.6)	3.1 ^1^ (±2.1)	289 ^1^ (±410)	4.5 ^1^ (±1.6)	6.1 ^1^ (±0.7)	2.9 ^1^ (±2.3)	285 ^1^ (±393)	4.6 ^1^ (±1.5)	5.8 ^1^ (±0.7)	2.9 ^1^ (±2.2)
>2	43 ^1^ (±41)	3.2 ^2^ (±1.2)	4.9 ^2^ (±0.5)	1.0 ^2^ (±1.8)	43 ^2^ (±39)	3.3 ^2^ (±1.0)	5.7 ^2^ (±0.4)	1.2 ^1^ (±1.5)	43 ^2^ (±40)	3.3 ^2^ (±1.1)	5.3 ^2^ (±0.6)	1.1 ^2^ (±1.6)
Fat/protein ratio												
≤1.1	465 ^1^ (±512)	5.2 ^1^ (±1.6)	5.7 ^1^ (±0.7)	3.8 ^1^ (±2.4)	443 ^1^ (±522)	4.9 ^1^ (±1.9)	6.2 ^1^ (±0.8)	3.4 ^1^ (±2.7)	454 ^1^ (±507)	5.0 ^1^ (±1.8)	5.9 ^1^ (±0.8)	3.6 ^1^ (±2.5)
1.1–1.4	265 ^2^ (±321)	4.7 ^2^ (±1.4)	5.4 ^1^ (±0.6)	3.2 ^2^ (±2.1)	291 ^2^ (±360)	4.7 ^2^ (±1.5)	6.2 ^1^ (±0.6)	3.1 ^1^ (±2.2)	278 ^2^ (±339)	4.7 ^1^ (±1.5)	5.8 ^1^ (±0.7)	3.2 ^1^ (±2.1)
>1.4	47 ^3^ (±39)	3.4 ^3^ (±1.1)	5.1 ^2^ (±0.5)	1.3 ^3^ (±1.6)	45 ^3^ (±41)	3.4 ^3^ (±1.0)	5.7 ^2^ (±0.4)	1.3 ^2^ (±1.4)	46 ^3^ (±40)	3.4 ^2^ (±1.1)	5.4 ^2^ (±0.6)	1.3 ^2^ (±1.5)
Milking type												
morning	217 ^1^ (±341)	4.3 ^1^ (±1.5)	5.3 ^1^ (±0.6)	2.6 ^1^ (±2.1)	216 ^1^ (±339)	4.2 ^1^ (±1.5)	5.9 ^1^ (±0.6)	2.4 ^1^ (±2.2)	217 ^1^ (±338)	4.3 ^1^ (±1.5)	5.6 ^1^ (±0.7)	2.5 ^1^ (±2.2)
evening	221 ^1^ (±319)	4.4 ^1^ (±1.5)	5.3 ^1^ (±0.6)	2.7 ^1^ (±2.2)	239 ^1^ (±361)	4.3 ^1^ (±1.6)	6.0 ^1^ (±0.6)	2.6 ^1^ (±2.2)	230 ^1^ (±339)	4.4 ^1^ (±1.5)	5.6 ^1^ (±0.7)	2.6 ^1^ (±2.2)
Average	219 ^A^ (±327)	4.3 ^A^ (±1.5)	5.3 ^A^ (±0.6)	2.6 ^A^ (±2.1)	229 ^A^ (±350)	4.3 ^A^ (±1.5)	5.9 ^B^ (±0.6)	2.5 ^A^ (±2.2)	224 (±338)	4.3 (±1.5)	5.6 (±0.7)	2.5 (±2.2)

Means in rows within somatic cell characteristics denoted with different letters differ significantly at *p* ≤ 0.05. Means in columns denoted with different numbers within factors differ significantly at *p* ≤ 0.05.

## Data Availability

The original contributions presented in the study are included in the article, further inquiries can be directed to the corresponding author.

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
