# Peer review of "The Influence of Selected Environmental Factors on the Number of Somatic Cells in Cistern and Alveolar Milk of Polish Holstein-Friesian Cows"

_animals, 2024, doi:10.3390/ani14152219_

Round 1

Reviewer 1 Report

Comments and Suggestions for Authors

It should be outlined why it would be useful to know whether SCC differences exist in alveolar and cisternal milk (fundamental research, biological knowledge, or could there be applications if there was a difference? 

A summary list of abbreviations should be shown at the beginning (or end) of the article. Some abbreviations are unconventional in the English speaking  literature and inconsistently used  (e.g. STB for fat protein ratio in line 193 , 231 and 233 and  and FPR in line 188).  SCC (Line 142,195) is used for somatic cell count, but LKS (L198) is also used. These need all be clearer. 

L 48 - reference 16, not 164

L 59 - these limits are legal limits for the milk quality above which milk is considered unfit for human consumption. SCC values below these limits do not guarantee healthy udders, these limits are usually set at 100,000-200,000 on an individual cow level. 

L 77 - Tancin's research is ref 40, not 41. (also ref 21).

Reference 34 - link does not work

L 153 how were the cows randomised? 15 cows out of a herd of 44. 

L 197 - correct to use logarithmic transformation, but linear score is more commonly used. The logic of the calculations is difficult to follow, especially including abbreviations which are unusual. What is the difference between LKS and SCC? 

L 236, Table 1: What does LSM stand for? How do you measure yield in the first and eights minute? 

L 309: Table 4 is very difficult to read, contains lots of irrelevant numbers and should be more summarized in the body of the text, could then be added as an appendix. 

L 319: STB here stands for energy protein balance in feed? The sentence should be re-written and clearer. 

The final part of the results session is actually part of the discussion. 

L 355 is not a sentence.

L 359 Sarikaya et al is ref 38, not 37

L 443 The actual SCC in 15 cows - average? average after linear transformation? 

The discussion should point out weaknesses and limitations, e.g. a small sample size from a single herd.

Reference 26: the link points to a different article from a different journal than the quoted paper (Iranian Journal of Veterinary Research). 

Overall the paper lacks clarity, is very difficult to read and it is not clear what the purpose of the research was. It does contain interesting findings (although none of them are novel), and a revised version (shorter and clearer) may be worth publishing. 

Comments on the Quality of English Language

Some sentences are not complete, see comments. 

Abbreviations are not always based on English terminology

Author Response

Response to review no. 1

Comments 1: It should be outlined why it would be useful to know whether SCC differences exist in alveolar and cisternal milk (fundamental research, biological knowledge, or could there be applications if there was a difference?

Response 1: Inflammation of cow udders is one of the most significant issues associated with the use of dairy cows worldwide. Its occurrence leads to a range of negative consequences related to milk production, the technological suitability of milk, and the longevity of cows. For these reasons, research on inflammation of the mammary gland should be considered very important from both an applied and a cognitive perspective. Their primary goals are usually to seek ways to reduce the causes of excessive somatic cell counts and the effects of their presence in milk. This study attempts to determine the location and time at which cows' bodies initiate an immune response. The question of whether alveolar milk or cisternal milk is mainly "responsible for the average somatic cell count in bulk milk" seems very intriguing. This issue, in addition to its cognitive significance, also has an applied aspect, which may, for example, concern whether by controlling certain phases of milking, the somatic cell count can be reduced, or whether intramammary antibiotic administration is aimed at addressing the "entire problem," etc.

Comments 2: A summary list of abbreviations should be shown at the beginning (or end) of the article. Some abbreviations are unconventional in the English speaking  literature and inconsistently used  (e.g. STB for fat protein ratio in line 193 , 231 and 233 and  and FPR in line 188).  SCC (Line 142,195) is used for somatic cell count, but LKS (L198) is also used. These need all be clearer.

Response 2: I agree with the reviewer's opinion. In the paper, the abbreviations FPR (Fat Protein Ratio) and SCC (Somatic Cell Count) have been standardized throughout all abbreviations and their derivatives. The corrections made according to the reviewer's suggestions have been marked in blue in the text of the paper. I believe that after this revision, particularly in the materials and methods section, the issue of the abbreviations used in the paper is now clear, and it is unnecessary to provide a detailed list of them in another part of the paper.

Comments 3: L 48 - reference 16, not 164

Response 3: I agree with the reviewer's opinion. The correction has been made in the text of the work.

Comments 4: L 59 - these limits are legal limits for the milk quality above which milk is considered unfit for human consumption. SCC values below these limits do not guarantee healthy udders, these limits are usually set at 100,000-200,000 on an individual cow level.

Response 4: I agree with the reviewer's suggestion, but SCC limits pertain to the assessment of milk quality at collection, not the evaluation of udder health status in cows.

Comments 5: L 77 - Tancin's research is ref 40, not 41. (also ref 21).

Response 5: I agree with the reviewer's opinion. The correction has been made in the text of the work.

Comments 6: Reference 34 - link does not work

Response 6: I disagree with the reviewer's suggestion - the link to this work is active.

Comments 7: L 153 how were the cows randomised? 15 cows out of a herd of 44.

Response 7: The number of cows included in the study was limited due to financial and organizational reasons. Such studies are labor-intensive and costly. Collecting milk samples at different stages of milking results in significant disruption to the milking process within the herd, affecting its organization and efficiency. For these reasons, the number of cows was restricted to 15. The study included 3 groups of animals, which, according to official assessments, were characterized by different levels of SCC in milk (>200, 201-400, and >400 thousand/mL).

Comments 8: L 197 - correct to use logarithmic transformation, but linear score is more commonly used. The logic of the calculations is difficult to follow, especially including abbreviations which are unusual. What is the difference between LKS and SCC?

Response 8: According to the response to Reviewer’s comment 2 – the abbreviations used in the paper have been standardized, and I believe the reviewer’s comment on this point is no longer relevant.

Comments 9: L 236, Table 1: What does LSM stand for? How do you measure yield in the first and eights minute?

Response 9: LSM stands for the Least Squares Mean. Milk yield at 1 and 8 minutes of milking was measured using a Bentley milk meter.

Comments 10: L 309: Table 4 is very difficult to read, contains lots of irrelevant numbers and should be more summarized in the body of the text, could then be added as an appendix.

Response 10: I only partially agree with the reviewer's opinion. The table does indeed contain a lot of data, but these are crucial elements for the results of the entire work, and it is difficult to omit them in a scientific paper.

Comments 11: L 319: STB here stands for energy protein balance in feed? The sentence should be re-written and clearer.

Response 11: In the paper, the sentence reads: "These data indicate that factors such as lactation period, cow age, milk yield, udder health status, milk let-down speed, and the level of energy-protein balance in feed doses (FPR) significantly and statistically differently differentiated SCC in cisternal and alveolar milk …". I propose leaving this sentence in its current form.

Comments 12: The final part of the results session is actually part of the discussion.

Response 12: I disagree with the reviewer's opinion.

Comments 13: L 355 is not a sentence.

Response 13: No comment.

Comments 14: L 359 Sarikaya et al is ref 38, not 37

Response 14: I agree with the reviewer's opinion. The correction has been made in the text of the work.

Comments 15: L 443 The actual SCC in 15 cows - average? average after linear transformation?

Response 15: I agree with the reviewer's opinion. I suggest adding the word "average" in the sentence "The average actual SCC....." on line 444.

Comments 16: The discussion should point out weaknesses and limitations, e.g. a small sample size from a single herd.

Response 16: I partially agree with the reviewer's opinion. Of course, a larger population would improve the reliability of the research results. However, the study presents data from 15 cows and 210 observations, which come from labor-intensive, costly, and challenging milk collection processes on the research farm. The cited studies by other authors on this topic were conducted with a similar or smaller number of cows – Sølverød et al. with 7 cows, Sarikaya et al. with 15 Simmental cows, 7 Brown Swiss cows, and 7 Holstein-Friesian cows, and Tančin et al. with 22 cows.

Comments 17: Reference 26: the link points to a different article from a different journal than the quoted paper (Iranian Journal of Veterinary Research).

Response 17: I agree with the reviewer's opinion. The correct link has been placed next to this literature entry.

Comments 18: Overall the paper lacks clarity, is very difficult to read and it is not clear what the purpose of the research was. It does contain interesting findings (although none of them are novel), and a revised version (shorter and clearer) may be worth publishing. 

Response 18: No comment.

Comments 19: Abbreviations are not always based on English terminology

Response 19: I agree with the reviewer's opinion. The abbreviations FPR (Fat Protein Ratio) and SCC (Somatic Cell Count) have been standardized throughout the text and their derivatives. Corrections made according to the reviewer's suggestions have been highlighted in blue in the text.

Reviewer 2 Report

Comments and Suggestions for Authors

In the study entitled “The influence of selected environmental factors on the number of somatic cells in cistern and alveolar milk of Polish Holstein-Friesian cows” the Authors purpose was to examine the variability in the number of somatic cells (SCC) in cisternal and alveolar milk of Polish Holstein-Friesian cows in relation to individual cow differences, lactation stage, age of cows, production level, milking speed, fat-to-protein ratio, and milking type. According to their findings, Authors concluded that SCC is characterized by high variability, influenced by a range of factors such as individual cow, lactation stage, age of cows, milking speed, and dietary fat-to-protein ratio. The influence of milking phase and milking type should be considered inconclusive based on the entire population studied. For half of the cows, SCC in cisternal milk was higher than in alveolar milk, while for the other half, the situation was reversed.

Please check the punctuation throughout the text as well as English language. On this regard, English language should be improved throughout the manuscript.

Specific Comments

Avoid the use of personal (i.e. us, our etc.) form throughout the text.

The title does not reflect the study. Indeed, Authors evaluated the variability in the number of somatic cells (SCC) in cisternal and alveolar milk of Polish Holstein-Friesian cows in relation to individual cow differences, lactation stage, age of cows, production level, milking speed, fat-to-protein ratio, and milking type. Cisternal and alveolar milk is not an environment factor Therefore, environmental is not an appropriate word. The title could be changed as “Variability of somatic cells number in cisternal and alveolar milk of Polish Holstein-Friesian cows”

Throughout the manuscript the word environmental should be avoided and/or replaced by another more appropriate.

The abstract adequately summarize methodology, results, and significance of the study. However, Authors should indicate the statistical analysis applied on the obtained data as well as the results together with P values should also indicated.

The introduction section sounds well and topic of the study well stated.

The Materials and Methods are well written and adequate information are reported. Authors should better indicate the time of milk sampling (hour of the day).

Did Authors evaluate the health status of enrolled animals? Overall, Authors should add more information on animals enrolled in the study.

Regarding statistical analysis, Did the Authors apply a normality test on data in order to assess their normal distribution? Please clarify this aspect as Authors used a parametric analysis.

Results and Discussion section are well written and the findings obtained in the study were well presented, discussed and justified with appropriate references. I have only the suggestion to make more harmonic the discussion section, it is somewhat redundant.

In the conclusion section Authors well summarize the results and the significance of the study. However I suggest to simplify it as it is too long. For example, the goal of the study has been previously reported.

I suggest to improve the quality of tables.

Comments on the Quality of English Language

Please check the punctuation throughout the text as well as English language. On this regard, English language should be improved throughout the manuscript.

Author Response

Response to review no. 2

Comments 1: Avoid the use of personal (i.e. us, our etc.) form throughout the text.

Response 1: I agree with the reviewer's suggestion. Appropriate corrections have been made to the text.

Comments 2: The title does not reflect the study. Indeed, Authors evaluated the variability in the number of somatic cells (SCC) in cisternal and alveolar milk of Polish Holstein-Friesian cows in relation to individual cow differences, lactation stage, age of cows, production level, milking speed, fat-to-protein ratio, and milking type. Cisternal and alveolar milk is not an environment factor Therefore, environmental is not an appropriate word. The title could be changed as “Variability of somatic cells number in cisternal and alveolar milk of Polish Holstein-Friesian cows”

Response 2: I partly agree with the reviewer's suggestion. After a detailed analysis, I think this suggestion can be accepted. Thus, the new title of the paper would be as follows: "Variability of Somatic Cell Count in Cisternal and Alveolar Milk of Polish Holstein-Friesian Cows".

Comments 3: Throughout the manuscript the word environmental should be avoided and/or replaced by another more appropriate.

Response 3: In animal science, factors affecting the variability of quantitative traits are traditionally divided into two main groups: genetic and environmental factors. Since one of the objectives of the study was to attempt to determine the influence of various factors such as lactation period, cow age, fat-to-protein ratio, production level, etc., on the number of somatic cells in milk, the term environmental factors was adopted as a common name. In the main part of the paper, this term was used 8 times, additionally in the abstract 3 times, and once in the simple summary. Since, as I believe, finding an appropriate equivalent of this term might be problematic and considering the fact that the term environmental factors is widely used in zootechnical literature, I propose keeping this term as suggested by the authors.

Comments 4: The abstract adequately summarize methodology, results, and significance of the study. However, Authors should indicate the statistical analysis applied on the obtained data as well as the results together with P values should also indicated.

Response 4: In the specialized literature, a whole range of presentations of variance analysis results for assessed traits is used. Presenting the results of the entire variance analysis for individual factors and traits is currently a rare practice in scientific papers. In the paper, it was decided to present the F-test with its significance at P≤0.05 and P≤0.01 (Table 2). In the authors' opinion, this provides sufficient information about the nature of the influence of individual factors on the traits analyzed in the paper.

Please maintain the method of presenting the variance analysis results as proposed by the authors.

Comments 5: The introduction section sounds well and topic of the study well stated.

Response 5: I agree with the reviewer's opinion.

Comments 6: The Materials and Methods are well written and adequate information are reported. Authors should better indicate the time of milk sampling (hour of the day).

Response 6: I agree with the reviewer's opinion. Furthermore, I propose to supplement the materials and methods section of the paper with the following information: milk samples were collected twice a day, at 6 a.m. and 6 p.m.

Comments 7: Did Authors evaluate the health status of enrolled animals? Overall, Authors should add more information on animals enrolled in the study.

Response 7: The authors did not assess the health status of the animals in the study. The animals involved in the study were fully healthy and showed no symptoms of metabolic, infectious, hoof, or udder diseases. Table 1 presents a range of data concerning the animals involved in the study, including their age structure, position in the production cycle, production level, metabolic status, and udder health status. I believe that this information satisfactorily characterizes the animals, thus fulfilling the reviewer's suggestion.

Comments 8: Regarding statistical analysis, Did the Authors apply a normality test on data in order to assess their normal distribution? Please clarify this aspect as Authors used a parametric analysis.

Response 8: In the paper, the authors adopted the well-known and widespread information that the number of somatic cells in milk does not follow a normal distribution. Therefore, a logarithmic transformation of the actual somatic cell count was conducted. All conclusions regarding the impact of factors on the number of somatic cells were based on the results after the logarithmic transformation.

Comments 9: Results and Discussion section are well written and the findings obtained in the study were well presented, discussed and justified with appropriate references. I have only the suggestion to make more harmonic the discussion section, it is somewhat redundant.

Response 9: I agree with the reviewer's opinion.

Comments 10: In the conclusion section Authors well summarize the results and the significance of the study. However I suggest to simplify it as it is too long. For example, the goal of the study has been previously reported.

Response 10: I agree with the reviewer's opinion. The summary chapter will be simplified, including omitting the research objectives.

Comments 11: I suggest to improve the quality of tables.

Response 11: I do not fully understand this suggestion from the reviewer. I assume it particularly concerns the extensive Table 4. There is a small problem with its construction and content. This table documents the key hypotheses analyzed in the paper concerning all the evaluated traits. Therefore, it is difficult to reduce its volume.

Comments 12: Please check the punctuation throughout the text as well as English language. On this regard, English language should be improved throughout the manuscript.

Response 12: I agree with the reviewer's opinion. I will ask the editorial team for a language correction of the text.

Round 2

Reviewer 1 Report

Comments and Suggestions for Authors

L 58: should add upper LEGAL limit, as it's not opinion but national law. 

Ref 34 - Link still does not work (may be a regional issue. )

L 153: the 15 cows are not random but selected according to given criteria which are outlined in the previous response. These selection criteria should be in the methods.

L 313-315: This is clearly an interpretation of the results, which belongs in the discussion. As comment 12 in the first report. The results section should not include interpretation but just state what the outcomes are. Same L 348-350 - whether something should be noted as interesting is discussion, not results.

L 461 and similar: it would be better to replace "our own" with "the authors' own" research.

Author Response

Response to Review Comments 1.1.

L 58: should add upper LEGAL limit, as it's not opinion but national law. 

I agree with the reviewer. The appropriate addition of "law" has been made to the text in line 58.

Ref 34 - Link still does not work (may be a regional issue. )

The link to this item is active for me.

L 153: the 15 cows are not random but selected according to given criteria which are outlined in the previous response. These selection criteria should be in the methods.

I agree with the reviewer's opinion. The materials and methods have been supplemented with the following sentence: "The study included 3 groups of animals, which, according to official assessments, were characterized by different levels of SCC in milk (>200, 201-400, and >400 thousand/mL".

L 313-315: This is clearly an interpretation of the results, which belongs in the discussion. As comment 12 in the first report. The results section should not include interpretation but just state what the outcomes are.

I think that the sentence "The results presented in Table 2 must be considered ambiguous and allow for the conclusion that SCC in cisternal and alveolar milk must be determined by factors other than milking phase" can be treated as a kind of textual connector between the content discussed in the results. Therefore, please leave it in its current version.

Same L 348-350 - whether something should be noted as interesting is discussion, not results.

I agree with the reviewer's suggestion. I propose removing the word "interesting" from the sentence.

L 461 and similar: it would be better to replace "our own" with "the authors' own" research.

I agree with the reviewer's suggestion. The appropriate correction has been made to the text.
